# Effects of Early Life Exposure to Sex Hormones on Neurochemical and Behavioral Responses to Psychostimulants in Adulthood: Implications in Drug Addiction

**DOI:** 10.3390/ijms23126575

**Published:** 2022-06-12

**Authors:** Maximiliano Elgueta-Reyes, Victoria B. Velásquez, Pedro Espinosa, Raúl Riquelme, Tatiana Dib, Nicole K. Sanguinetti, Angélica P. Escobar, Jonathan Martínez-Pinto, Georgina M. Renard, Ramón Sotomayor-Zárate

**Affiliations:** 1Laboratorio de Neuroquímica y Neurofarmacología, Centro de Neurobiología y Fisiopatología Integrativa (CENFI), Instituto de Fisiología, Facultad de Ciencias, Universidad de Valparaíso, Valparaíso 2360102, Chile; max.elgueta17@gmail.com (M.E.-R.); victoria.velasquezp@gmail.com (V.B.V.); pedro.espinosa.qf@gmail.com (P.E.); raulriquelmen@gmail.com (R.R.); tatiana.dib@alumnos.uv.cl (T.D.); nicole.sanguinetti@gmail.com (N.K.S.); angelica.escobar@uv.cl (A.P.E.); jonathan.martinez@uv.cl (J.M.-P.); 2Centro de Investigación Biomédica y Aplicada (CIBAP), Escuela de Medicina, Facultad de Ciencias Médicas, Universidad de Santiago de Chile, Santiago 9170022, Chile

**Keywords:** dopamine, sex hormones, accumbens, striatum, amphetamine, FSCV, microdialysis

## Abstract

Early life exposure to sex hormones affects several brain areas involved in regulating locomotor and motivation behaviors. Our group has shown that neonatal exposure to testosterone propionate (TP) or estradiol valerate (EV) affected the brain dopamine (DA) system in adulthood. Here, we studied the long-lasting effects of neonatal exposure to sex hormones on behavioral and neurochemical responses to amphetamine (AMPH) and methylphenidate (MPD). Our results show that AMPH-induced locomotor activity was higher in female than male control rats. The conditioned place preference (CPP) to AMPH was only observed in EV male rats. In EV female rats, AMPH did not increase locomotor activity, but MPD-induced CPP was observed in control, EV and TP female rats. Using in vivo brain microdialysis, we observed that AMPH-induced extracellular DA levels were lower in nucleus accumbens (NAcc) of EV and TP female rats than control rats. In addition, MPD did not increase NAcc extracellular DA levels in EV rats. Using in vivo fast-scan cyclic voltammetry in striatum, MPD-induced DA reuptake was higher in EV than control rats. In summary, our results show that early life exposure to sex hormones modulates mesolimbic and nigrostriatal DA neurons producing opposite neurochemical effects induced by psychostimulant drugs in NAcc or striatum.

## 1. Introduction

The brain is particularly sensitive to sex hormones, which support working memory, learning, sensory integration, homeostatic regulation, mood, emotion and motivation, among others [1,2,3]. Sex hormone receptors are expressed in numerous brain areas, including the prefrontal cortex (PFC), amygdala, hippocampus, locus coeruleus and hypothalamus [1,3]. Another important brain circuit that expresses sex hormone receptors is the mesocorticolimbic system [4,5] formed by dopaminergic neurons from the ventral tegmental area (VTA) that project their axons to nucleus accumbens (NAcc) and PFC [6,7]. This circuit is activated by natural rewards, such as sex [8] and food [9], as well as drugs of abuse [10], which also increase dopamine (DA) release in NAcc and striatum [11]. Psychostimulant drugs, such as amphetamine (AMPH), cocaine and methylphenidate (MPD), inhibit and block the DA transporter (DAT), respectively, preventing DA uptake and increasing extracellular NAcc DA levels, which favors addictive-like behaviors [12].

Importantly, sex plays a critical role in drug addiction. It has been shown that female rats have higher cocaine choice than males, where ovariectomization (OVX) reduces the cocaine choice [13]. Additionally, locomotor activity induced by AMPH is higher in female than male rats [14]. Accordingly, it has been shown that male rats have higher expression levels of DAT [15]. In addition, it is important to mention that the activities of DAT and the vesicular monoamine transporter 2 (VMAT2) are different during the estrous cycle stages [16]. In previous works, we have shown an increase in DA tissue levels in different areas of the brain, including the reward system, in neonatal EV-treated rats [17,18,19,20]. In accordance with these results, we observed an increase in tyrosine hydroxylase (TH) expression in VTA and substantia nigra in adult rats [20]. Moreover, neonatal exposure to EV increased induced morphine rewarding behaviors in female rats, showing at electrophysiological levels an increase in the frequency of spontaneous excitatory postsynaptic currents in NAcc medium spiny neurons (MSNs) [17].

We also observed that adult female rats treated with TP have lower MPD-induced locomotor activity compared to control and EV-treated rats and that this reduction in locomotor activity is related to lower NAcc DAT expression [18]. 

However, the long-lasting effects of neonatal exposure to sex hormones on midbrain dopaminergic neurons in response to psychostimulant drugs have not been thoroughly studied. Here, we examined the effects of EV or TP exposure during the first hours of postnatal life on locomotor activity and conditioned place preference induced by MPD (5 mg/kg i.p.) or AMPH (1 mg/kg i.p.) in female and male adult rats. Tonic and phasic DA release in the NAcc and striatum of rats treated with these psychostimulants was assessed by microdialysis and fast-scan cyclic voltammetry.

## 2. Results

### 2.1. Conditional Place Preference (CPP)

Administration of AMPH and MPD did not produce statistically significant differences (Figure 1A) (interaction: F_(4, 57)_ = 1.671; *p* = 0.1692; neonatal treatment: F_(2, 57)_ = 0.3240; *p* = 0.7246; drug: F_(2, 57)_ = 21.24; *p* < 0.0001) in male rats. However, in female rats, we observed an effective conditioning to the rewarding effects of AMPH and MPD in control rats, which was not observed in EV and TP female rats (Figure 1B) (interaction: F_(4, 60)_ = 4.686; *p* = 0.0023; neonatal treatment: F_(2, 60)_ = 4.775; *p* = 0.0119; drug: F_(2, 60)_ = 26.17; *p* < 0.0001). 

### 2.2. Locomotor Activity

Cumulative basal locomotor activity (0–30 min) did not differ between control versus EV and TP male rats. However, we only observed a significant difference between EV versus TP male rats (Figure 2B) (F_(2, 30)_ = 4.854; *p* = 0.0149). On the other hand, cumulative basal locomotor activity was not different between control, EV and TP female rats (Figure 2D) (F_(2, 34)_ = 1.150; *p* = 0.3286). 

With regard to cumulative locomotor activity induced by AMPH, we did not observe statistically significant differences between control, EV and TP male rats (Figure 2B) (F_(2, 30)_ = 0.5718; *p* = 0.5705). However, in EV female rats, we only observed a significant reduction in AMPH-induced locomotor activity compared to control female rats (Figure 2D) (F_(2, 34)_ = 4.425; *p* = 0.0196). Interestingly, we observed a sex difference in AMPH-induced locomotor activity between female and male control rats (Figure 2E) (*p* = 0.0094).

### 2.3. In Vivo Brain Microdialysis

Systemic administration of AMPH increases DA release in NAcc with regard to baseline levels in control, EV or TP male rats. However, we did not observe significant differences in the magnitude of DA levels induced by AMPH between control, EV and TP male rats (Figure 3A) (interaction: F_(14, 72)_ = 0.3932; *p* = 0.9727; time: F_(7, 72)_ = 15.40; *p* < 0.0001; neonatal treatment: F_(2, 72)_ = 2.372; *p* = 0.1006). However, in adult female rats exposed to sex hormones during the first hours of postnatal life, AMPH-induced increases in DA levels in NAcc were lower than female control rats (Figure 3C) (interaction: F_(14, 104)_ = 2.446; *p* = 0.0051; time: F_(7, 104)_ = 13.12; *p* < 0.0001; neonatal treatment: F_(2, 104)_ = 14.88; *p* < 0.0001). On the other hand, when comparing the effect of MPD in control, EV and TP male rats, we only observed significant differences in neonatal treatment (EV) and in the time factor (Figure 3B) (interaction: F_(14, 104)_ = 1.487; *p* = 0.1287; time: F_(7, 104)_ = 7.098; *p* < 0.0001; neonatal treatment: F_(2, 104)_ = 9.963; *p* = 0.0001). Interestingly, the neurochemical effect of MPD in adult female rats was different depending on the type of sex hormone injected during the first hours of postnatal life. In this sense, neonatal exposure to TP increases, while exposure to EV reduces MPD-induced DA extracellular levels in NAcc (Figure 3D) (interaction: F_(14, 112)_ = 5.174; *p* < 0.0001; time: F_(7, 112)_ = 17.24; *p* < 0.0001; neonatal treatment: F_(2, 112)_ = 32.43; *p* < 0.0001).

### 2.4. In Vivo Fast-Scan Cyclic Voltammetry (FSCV)

In vivo FSCV was performed in striatum of anesthetized rats. Midbrain electric stimulation was carried out every 5 min to collect baseline values—3 measures after saline injection (1 mL/kg i.p.) and 15 measures after MPD administration (5.0 mg/kg i.p.). With regard to striatal DA peak height and tau, we observed a statistically significant effect for neonatal treatment and time in male (Figure 4A, (interaction: F_(40, 294)_ = 0.9712; *p* = 0.5245; time: F_(20, 294)_ = 11.09; *p* < 0.0001; neonatal treatment: F_(2, 294)_ = 28.13; *p* < 0.0001)), Figure 4B, (interaction: F_(40, 293)_ = 0.9027; *p* = 0.6415; time: F_(20, 293)_ = 9.903; *p* < 0.0001; neonatal treatment: F_(2, 293)_ = 24.24; *p* < 0.0001))) and female rats (Figure 4E, (interaction: F_(40, 273)_ = 0.7017; *p* = 0.9119; time: F_(20, 273)_ = 20.57; *p* < 0.0001; neonatal treatment: F_(2, 273)_ = 30.83; *p* < 0.0001)), Figure 4F, (interaction: F_(40, 271)_ = 1.085; *p* = 0.3441; time: F_(20, 271)_ = 14.97; *p* < 0.0001; neonatal treatment: F_(2, 271)_ = 30.03; *p* < 0.0001))).

### 2.5. DAT mRNA Expression in VTA and SN 

DAT mRNA expression was measured in the VTA and SN of adult male and female rats of experimental groups. VTA and SN are the main brain areas involved in synthesizing DA for release in NAcc and striatum, respectively. The DAT mRNA expression in VTA was lower in TP male rats than control and EV male rats. Meanwhile, in female rats, the DAT mRNA expression in VTA was lower in animals exposed to sex hormones than control female rats Figure 5A, (interaction: F_(2, 36)_ = 5.044; *p* = 0.0117; sex: F_(1, 36)_ = 0.04849; *p* = 0.8269; neonatal treatment: F_(2, 36)_ = 9.407; *p* = 0.0005). Interestingly, DAT mRNA expression in SN was only increased in EV male rats with regard to the control group. However, in the SN of female rats, we did not observe significant differences in DAT mRNA expression Figure 5B, (interaction: F_(2, 23)_ = 30.24; *p* < 0.0001; sex: F_(1, 23)_ = 58.92; *p* < 0.0001; neonatal treatment: F_(2, 23)_ = 18.11; *p* < 0.0001).

## 3. Discussion

The purpose of this new work was to elucidate the effects of deleterious early life events on the long-term neurochemical and behavioral alterations induced by psychostimulant drugs. In this context, our model of neonatal exposure to sex hormones proved to be a reprograming model for dopaminergic neurons, promoting the development of neuropsychiatric disorders, such as addiction to drugs of abuse. Here, we studied whether a single neonatal dose of EV or TP modulates the effects of psychostimulant drugs, such as AMPH and MPD, in mature brain circuits associated with motivation, locomotion and pathologies, such as addiction or attention deficit hyperactivity disorder (ADHD), among others [19,20]. In addition, our interest in understanding the sex differences observed in mesolimbic and nigrostriatal pathways and the international initiatives that encourage the scientific community to consider the sex of animals in preclinical studies [21,22] led us to compare the effects of EV and TP on psychostimulant actions in both females and males. 

At the behavioral level, repeated AMPH administration produced CPP in control rats and EV male rats, without effects on TP male rats or EV and TP female rats. This behavioral effect is consistent with the locomotor effects observed with a single dose of AMPH in females, since in EV female rats, the AMPH-induced locomotor activity was abolished and tended to be less in female TP rats. In this context, AMPH is an old drug with an abuse potential, and it has long been studied [23,24,25]. At the pharmacological level, AMPH is considered a DAT substrate, promoting NAcc and striatum DA release into the synaptic cleft [11,25,26,27]. Considering this mechanism of action, our group studied the effects of early exposure to sex hormones on the expression of key proteins for DA neurotransmission in adulthood. Previously, we observed that neonatal exposure to TP significantly decreases the glycosylated DAT expression in NAcc and MPD-induced locomotor activity in adult female rats [18]. In this sense, neonatal exposure to bisphenol A (an environmental pollutant with estrogenic activity) also decreases striatal DAT expression in adult rodents [28,29]. At the neurochemical level, we observed that neonatal exposure to EV or TP decreases AMPH-induced DA release in the NAcc of adult female rats, suggesting that the main pharmacological target of AMPH could have decreased expression. Supporting this hypothesis, we observed that DAT mRNA expression in VTA decreased significantly in EV and TP female rats, and this was consistent with the decrease in AMPH-induced behavioral effects. An interesting question is whether neonatal exposure to sex hormones produces long-lasting changes in these brain pathways or whether there are changes in plasma levels of sex hormones in adulthood that affect the DAT expression, since ovariectomized adult rats have lower NAcc DAT expression than control adult rats [30]. In this context, we observed that neonatal exposure to EV does not affect estradiol plasma levels in adulthood [31]; therefore, early EV exposure is likely to produce epigenetic changes that lead to an altered pattern of gene expression in adult life, such as we have shown previously [32].

MPD is considered to be a DAT blocker, which induces a cumulative increase in DA extracellular levels [19,33,34]. In this work, MPD administration for 5 days produced significant CPP in both male and female control rats; however, conditioning was observed only in EV and TP female rats. In a previous work, we observed that MPD-induced locomotor activity did not differ between control, EV and TP male rats. However, in females, MPD induced an increase in locomotor activity in control and EV rats, whereas this increase was lower in TP female rats [18]. These results were associated with a decrease in NAcc protein levels of glycosylated DAT. At the neurochemical level, MPD effects were evaluated by NAcc brain microdialysis and striatal FSCV. In NAcc, the neonatal exposure to EV decreased extracellular DA levels induced by MPD in male and female rats, while TP increased DA levels. In striatum, the neonatal exposure to EV increased DA peak height and *decay time* constant (τ) in male and female rats. However, neonatal exposure to TP only increased striatal DA peak height in female rats. Interestingly, these neurochemical findings are correlated with increased DAT mRNA expression in SN for EV male rats and suggest that the supraphysiological administration of sex hormones during critical periods of development affects both tonic and phasic DA levels in a region-specific manner.

## 4. Materials and Methods

### 4.1. Reagents

EV, TP, sesame oil, DA standard, EDTA and 1-octanesulfonic acid were purchased from Sigma-Aldrich, Inc. (St. Louis, MO, USA). AMPH sulfate was obtained from Laboratorio Chile S.A. (ISPCH N° F-1386/18, Ñuñoa, Santiago, Chile), and MPD was obtained from Laboratorio Andrómaco S.A. (ISPCH N° F-7543/21, Peñalolén, Santiago, Chile). All other reagents were of analytical and molecular grade. The doses of AMPH and MPD were selected from publications where behaviors such as CPP, locomotor activity and in vivo brain microdialysis were performed [19,35].

### 4.2. Animals

In total, 146 male and 159 female Sprague Dawley rats from different litters were considered for the following experimental groups: female control (*n* = 51), male control (*n* = 45), female EV (*n* = 54), male EV (*n* = 56), female TP (*n* = 47) and male TP (*n* = 52). Animals from the vivarium of the Faculty of Science of the Universidad de Valparaíso were used and housed in a temperature- and humidity-controlled room (22 ± 2 °C; 50 ± 5%, respectively) under artificial illumination (12 h light/12 h dark; light on at 08:00 a.m.), with food and water *ad libitum*. Efforts were made to minimize the number of rats used and their suffering. 

### 4.3. Experimental Procedure

Animals of both sexes were injected during the first hours of postnatal (PN) life with a single dose of EV (0.1 mg/50 µL of sesame oil s.c.), TP (1 mg/50 µL of sesame oil s.c.) or vehicle (control group: 50 µL of sesame oil s.c.). Sex hormone doses were previously published [17,18,32,36], demonstrating the estrous cycle to be in a permanently diestrus phase [37]. Female control rats were used between PN days (PND) 60 to 62 to maintain the same phase of the estrous cycle. Experimental groups were randomly assigned for the following experimental protocols.

### 4.4. Behavioral Studies

In total, 205 rats were used for CPP (*n* = 135) and locomotor activity (*n* = 70) experiments.

#### 4.4.1. Conditional Place Preference (CPP)

Rats used for CPP were assigned to the following experimental groups: female control (*n* = 22), male control (*n* = 21), female EV (*n* = 26), male EV (*n* = 25), female TP (*n* = 21) and male TP (*n* = 20). The CPP apparatus and the experimental protocol were previously used [17,36]. Briefly, the CPP protocol consisted of three parts: pre-test (one day before the conditioning period), the conditioning period and the test (24 h after the last injection). The conditioning period was associated with the psychostimulants-induced reward in the white compartment for 60 min, where we used AMPH (1 mg/kg i.p.) or MPD (5 mg/kg i.p.) in the morning and saline injection (1 mL/kg i.p.) in the afternoon (black compartment). The control group received the saline injection in the morning and the afternoon in the black and white compartment, respectively. Data are represented as time difference (ΔT) in seconds between the time spent in the white compartment on the test day and the pre-test day.

#### 4.4.2. Locomotor Activity

Rats used for measuring AMPH-induced locomotor activity were assigned to the following experimental groups: female control (*n* = 12), male control (*n* = 9), female EV (*n* = 11), male EV (*n* = 14), female TP (*n* = 14) and male TP (*n* = 10). The experimental protocol was previously published [31]. Briefly, basal locomotor activity was measured for the first 30 min. At 30 min, AMPH (1 mg/kg i.p.) was injected, and locomotor activity was recorded for 60 min. Videos were analyzed using ANY-Maze™ video tracking system (Stoelting™ Co., Wood Dale, IL, USA), measuring the total distance traveled (m) every 5 min. 

### 4.5. Neurochemical Studies

In total, 28 and 33 rats were used for in vivo brain microdialysis and in vivo fast-scan cyclic voltammetry experiments, respectively.

#### 4.5.1. In Vivo Brain Microdialysis

Microdialysis experiments were performed on anesthetized rats, following a previously published protocol [36]. Briefly, a microdialysis probe (2 mm membrane length, model MAB 2.14.2, 35,000 Da cut-off, Microbiotech AB, Stockholm, Sweden) was implanted in the NAcc (anteroposterior: +1.5 mm; mediolateral: +1.5 mm; dorsoventral: −7.8 mm). Microdialysis probe was perfused with artificial cerebrospinal fluid solution (aCSF) at a flow rate of 2 μL/min using an infusion pump (model RWD 210, RWD Life Science Co. Ltd., Shenzhen, China). After a stabilization period of 90 min, dialysate samples were collected after 20 min each. Two baseline samples (0–40 min) were collected in 3 μL of 0.2 M perchloric acid. At 40 min, AMPH (1 mg/kg i.p.) or MPD (5 mg/kg i.p.) was injected, and 6 dialysate samples were collected between 40 and 160 min. Once the experiment was finished, the animal was euthanized, and the brains were removed to verify the location of microdialysis probe.

Dialysates samples were injected into an HPLC system coupled with electrochemical detection [36], and for chromatographic separation, a C-18 reverse phase column (model Kromasil 100-3.5-C18, AkzoNobel, Bohus, Sweden) was used. The column was pumped at a flow rate of 0.2 mL/min using the mobile phase (0.625 mM 1-octanesulfonic acid, 0.1 M NaH_2_PO_4_, 1 mM EDTA, 1.0% tetrahydrofuran, 6% CH_3_CN and adjusted pH to 2.4). The quantification was derived comparing the peak area and elution time with reference standards, using the Program ChromPass (Jasco Co. Ltd., Tokyo, Japan).

#### 4.5.2. In Vivo Fast-Scan Cyclic Voltammetry

FSCV experiments were performed on anesthetized rats, following a previously published protocol [38]. Briefly, 3 electrodes (working, reference and bipolar stimulating electrode) were implanted using the coordinates from the Rat Brain Atlas. A working electrode (glassy-carbon microelectrode) was implanted in striatum (anteroposterior: +1.3 mm; mediolateral: +2.5 mm; dorsoventral: −4.0 mm). The reference electrode (Ag/AgCl) was implanted in the contralateral cortex, and the bipolar stimulating electrode (model MS 303/2A, Plastics one Inc., Roanoke, VA, USA) was implanted in the midbrain (anteroposterior: −5.2 mm; mediolateral: +1.3 mm; dorsoventral: −7.5 mm).

The waveform of FSCV potential for working electrode was −0.4 to 1.2 V and back to −0.4 V (Ag/AgCl). The scan rate of 400 V/s was assessed every 100 ms using a voltammeter/amperometer (model Chem-Clamp Potensiostat, Dagan Corporation, Minneapolis, MN, USA). The stimulation of phasic DA release was performed under the following parameters: monophasic +, 60 pulses, 60 Hz, 4 ms, 300 μA (current stimulus isolator NL800A; Digitimer, Ltd., Hertfordshire, UK). Demon Voltammetry and Analysis software (Wake Forest Health Sciences, Winston-Salem, NC, USA) were used [39]. Striatal phasic DA release was stimulated every 5 min according to the following protocol: first, three steady baseline records; second, 3 records after saline injection (1 mL/kg, i.p.); and third, 15 records after MPD injection (5 mg/kg i.p.). After each experiment, working electrodes were calibrated using aCSF containing 1 μM dopamine. Demon Voltammetry allows the measurement of kinetic parameters, such as Tau, which was determined from the “exponential fit curves based on peak cursor and post-stim cursor positions using a least squares constrained exponential fit algorithm (National Instruments)” [39]. Tau and other kinetic parameters are positively correlated with changes of the Michaelis constant (Km), suggesting that these are accurate measures of DA uptake [39].

### 4.6. qRT-PCR

Real-time PCR was used to determine whether the DAT mRNA expression changed in NAcc of adult female and male rats exposed to sex hormones at PND1. Briefly, total RNA was extracted using the E.Z.N.A.^®^ Total RNA Kit I (Cat. N° R6834-02; Omega Biotek, Norcross, GA, USA), and it was quantified using the microplate Spectrophotometer Epoch (BioTek Inc., Winooski, VT, USA). RNA integrity was checked through agarose gel electrophoresis, and the reverse transcription was performed using the PrimeScript RT reagent Kit (Cat. N° RR047A; TaKaRa, Bio Inc., San Jose, CA, USA), according to the manufacturer’s instructions. Real-time PCR was performed using the QuantiTect SYBR Green PCR Kit (No. 204143, Qiagen, Valencia, CA, USA), in accordance with the manufacturer’s instructions. For specific gene amplification, a standard protocol of 40 cycles was used in a CFX96™ Real-Time System (Bio-Rad, Hercules, CA, USA). The amplification conditions included an initial polymerase activation at 95 °C for 10 min, followed by 40 cycles of cDNA amplification phases: 95 °C for 10 s for denaturation; 63.8 °C for 20 s (DAT) or 65 for 15 s (18S) for annealing; and 72 °C for 25 s (DAT) or 30 s (18S) for extension. The DAT primer was designed from data published in GenBank, access N° NM_012694.2, DAT forward 5′-CTTGGCATTGTCCTGGCTACTTTCC-3′ and DAT reverse 5′-CAGCATAGCCGCCAGTACAGGTTG-3′. To normalize DAT mRNA content, ribosomal 18S mRNA was measured in each protocol using primers that have been reported previously [40]: 18S forward, 5′-TCAAGAACGAAAGTCGGAGG-3′ and 18S reverse 5′-GGACATCTAAGGGCATCACA-3′ (GenBank access N° NR_046237.2). Specificity of the generated amplicons was confirmed by performing melting curves at the end of each reaction. Results were expressed as fold change by the 2^ΔΔCT^ method [41].

### 4.7. Statistical Analysis

Data were expressed as mean ± SEM. Two-way analysis of variance (ANOVA) multiple comparisons followed by Tukey’s post hoc test were performed to analyze Figure 1, Figure 3, Figure 4 and Figure 5. One-way ANOVA multiple comparisons followed by Tukey’s post hoc test were performed to analyze Figure 2B,D, and two-tailed unpaired *t*-test was performed to analyze Figure 2E. The statistical analyses were carried out with GraphPad Prism v 9.3.1 (GraphPad Software, San Diego, CA, USA), and *p* < 0.05 was considered statistically significant. 

## 5. Conclusions

The results obtained based on our model of neonatal reprograming with sexual hormones indicate that females treated with EV display decreased behavioral responses to the rewarding effects of psychostimulants, which is consistent with lower DA release in NAcc. Additionally, we find a decrease in the expression of DAT mRNA, which may explain the reduced effects of psychostimulants. Interestingly, several studies show that estrogens increase the rewarding effects of drugs [42]; however, our results show that neonatal EV administration produces the opposite effect. In males, the neonatal administration of sex hormones seems to have minimal effects on responses to psychostimulants in adulthood. Future studies must elucidate the mechanisms underlying the different actions of estrogens at different neurodevelopment stages.

## Figures and Tables

**Figure 1 ijms-23-06575-f001:**
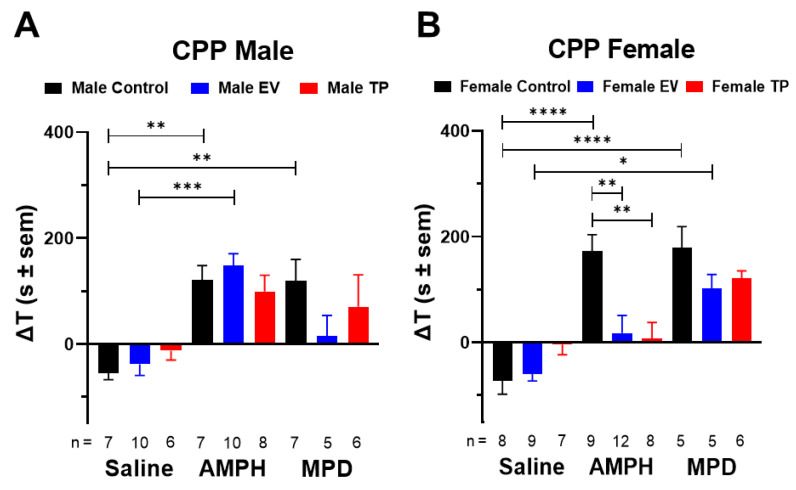
Effects of exposure to estradiol valerate (EV) and testosterone propionate (TP) at postnatal day (PND) 1 on amphetamine (AMPH) and methylphenidate (MPD) induced conditioned place preference (CPP) in adult male (panel **A**) and female (panel **B**) rats. Data are shown as the time difference (ΔT) in seconds between the time spent in the white drug-paired compartment on the test day and on the pre-test day. Data are expressed as means ± SEM (standard error mean), and the statistical analysis used was two-way ANOVA (* *p* < 0.05; ** *p* < 0.01; *** *p* < 0.001 and **** *p* < 0.0001).

**Figure 2 ijms-23-06575-f002:**
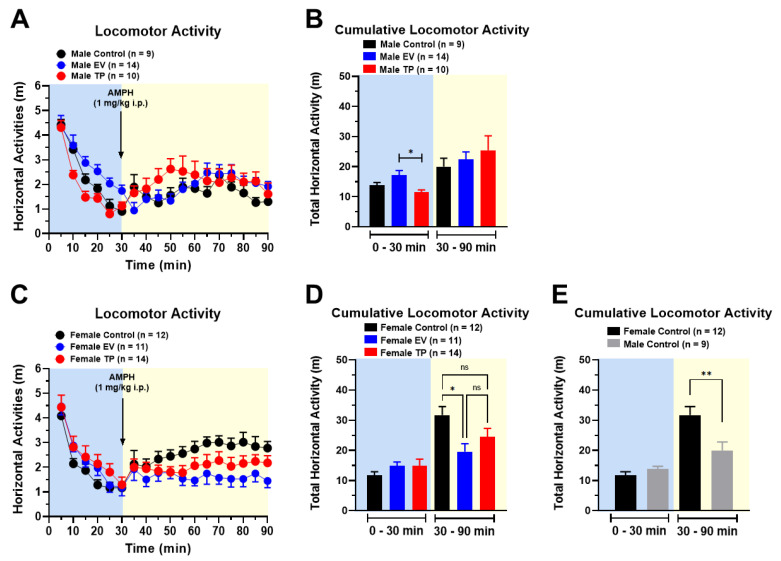
Basal and AMPH-induced locomotor activity (1 mg/kg i.p.) in male and female rats neonatally treated with saline, EV and TP. In panels (**A**,**C**), the time course of basal and AMPH-induced locomotor activity is shown, expressed as distance traveled (m). Panels (**B**,**D**) show cumulative basal (0–30 min) and AMPH-induced (30–90 min) locomotor activity in male and female rats. Panel (**E**) compares the cumulative locomotor activity of control males and females, neonatally treated with saline. Data are expressed as mean ± SEM (mean standard error), and the statistical analysis used was one-way ANOVA (* *p* < 0.05; ** *p* < 0.01).

**Figure 3 ijms-23-06575-f003:**
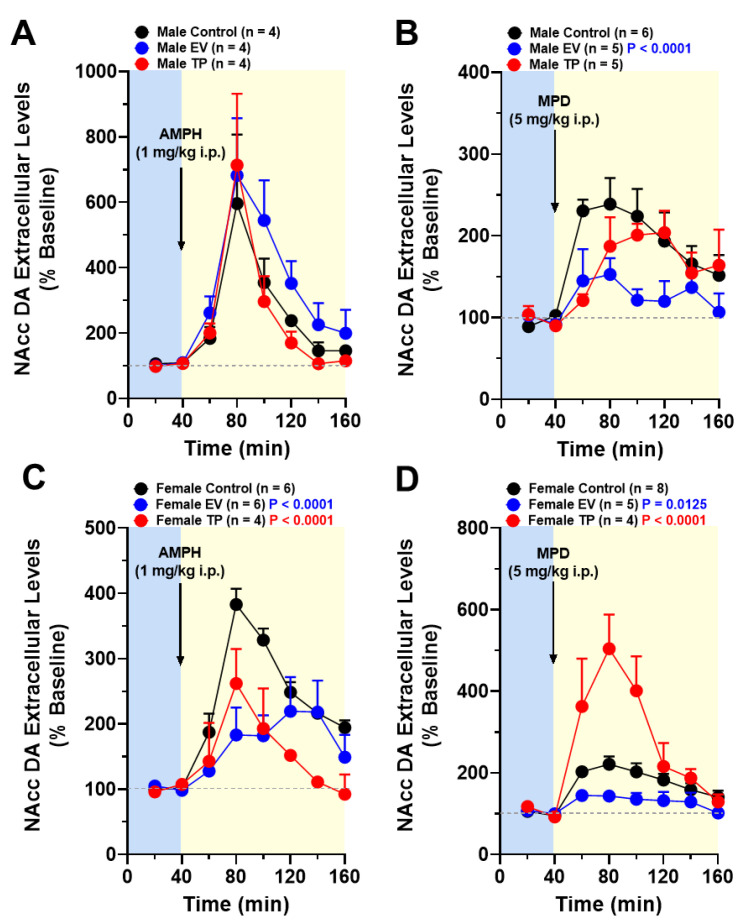
Extracellular levels of dopamine (DA) in nucleus accumbens (NAcc) following systemic administration of AMPH and MPD. Graphs represent the time course of DA release in NAcc expressed as the mean % basal DA release (mean ± SEM) in male (**A**,**B**) and female (**C**,**D**) rats neonatally treated with estradiol valerate (EV) or testosterone propionate (TP). Statistical analysis to compare DA release time course curves was performed using a two-way analysis of variance ANOVA.

**Figure 4 ijms-23-06575-f004:**
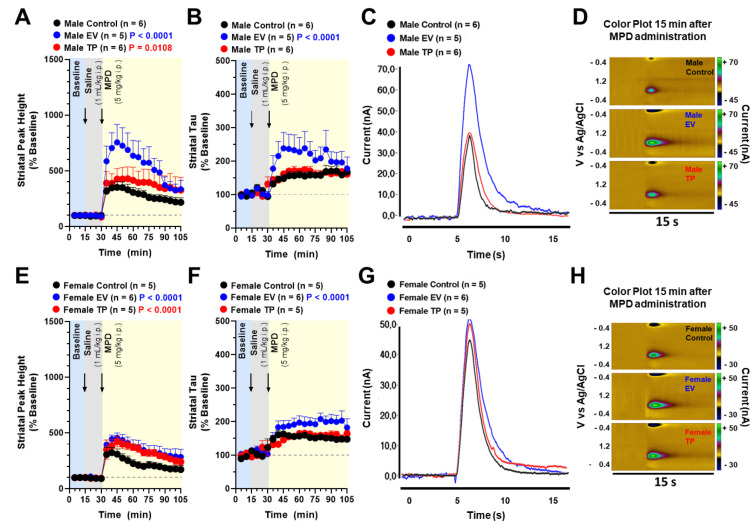
DA release and uptake dynamics obtained in DS before and after the acute i.p. MPH administration. Graphs represent the time course of electrically evoked DA release expressed as percent of the baseline shown as mean ± SEM for male (**A**) and female (**E**) rats neonatally treated with estradiol valerate (EV) or testosterone propionate (TP). Clearance for phasic DA (Tau) was measured under the same conditions for male (**B**) and female (**F**) rats. Example evoked DA traces in response to MPD administration for male (**C**) and female (**G**) rats. Color plot representations of cyclic voltametric data collected over 15 s for male (**D**) and female (**H**) rats. The current generated by oxidation/reduction is depicted in color in response to the potential applied and the time of collection. Under these conditions, the oxidation of dopamine occurs at 0.6 V. Data were analyzed by two-way ANOVA.

**Figure 5 ijms-23-06575-f005:**
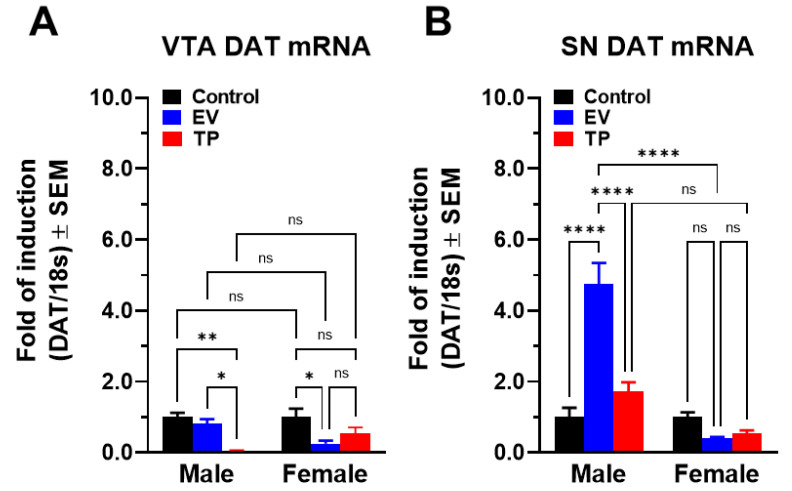
DAT mRNA expression in ventral tegmental area (VTA) (**A**) and substantia nigra (SN) (**B**) of male and female rats neonatally treated with estradiol valerate (EV) or testosterone propionate (TP). All data are normalized for 18S expression levels within the same sample. Results are expressed as fold induction relative to control group and represent the mean ± SEM. Data were analyzed by two-way ANOVA (* *p* < 0.05; ** *p* < 0. 01; **** *p* < 0.0001).

## Data Availability

All datasets generated for this study are included in the manuscript.

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
