# Peer review of "Effects of Early Life Exposure to Sex Hormones on Neurochemical and Behavioral Responses to Psychostimulants in Adulthood: Implications in Drug Addiction"

_ijms, 2022, doi:10.3390/ijms23126575_

Round 1
Reviewer 1 Report
The manuscript by Elgueta-Reyes et al titled “Effects of Early-Life Exposure to Sex Hormones on 2 Neurochemical and Behavioral Responses to Psychostimulants 3 in the Adulthood: Implications in Drug Addiction” In the present study they demonstrated that early-life exposure to sex hormones In modulates mesolimbic and In nigrostriatal DA neurons 26 producing opposite neurochemical effects induced by psychostimulant drugs in striatum. The experiments are technically sound, and the data is strongly convincing. However, I have one major concern that needs to be addressed before this work is suitable for publication.
- In the study, the author has used the RT-PCR to measure DAT mRNA levels in VTA and SNc and normalized them to 18S expression levels. What is the rationale behind using the 18S expression level to normalize? It would be ideal to use TH. TH mainly represents dopaminergic neurons. I would like to see how the data looks when normalized to TH. Also, it would be ideal to see the effect on protein levels.
- Amphetamine is known to down-regulate DAT in the NAC. It would be ideal to see the effect of early-life exposure to sex hormones in response to amphetamine in this region.
- There are many typos in regard when writing p-value. For example, the p-value is written as p=0, Please correct it.
Reviewer 2 Report
Authors
I have carefully read the manuscript of Elgueta-Reyes et al. on the effects of neonatal exposure to sex hormones. The work is very interesting and well planned. The study fits into the international initiatives that promotes the study of gender differences in the effect of drugs. The results are remarkable and experimental methods are suitable.
Authors’ hypothesis starts in the sense that a single neonatal dose of EV or TP could modulate the effects of psychostimulant drugs such as AMPH and MPD in mature brain circuits. However, all the effects investigated were no assessed by both drugs. Briefly:
- CPP: AMPH and MPD
- Locomotor activity (LA): AMPH
- In vivo microdialysis: AMPH and MPD
- In vivo fast scan cyclic voltammetry: MPD
I can understand that the effects of sex hormones in LA of MPD were investigated in a previous paper, but i cannot understand the reason to avoid the study of fast scan cyclic voltammetry with AMPH.
Authors should include the reason for the AMPH and MPD chosen doses. Probably the paper could improve and find more differences if several doses of each drug were tested
My main objections to the work are the conclusions that are obtained from the results from the statistical analysis
Probably authors expect a difference of the hormone effects between female and male adult rats because they stated in section 4.7. Statistical Analysis: “Two-way ANOVAs followed by Newman–Keuls post hoc tests were performed for all experiments to determine a significant interaction of treatment by sex”. It should be noted that they have almost never done an ANOVA with gender as a variable, except for the results in the figure 2E.
Results section 2.1 CPP
In Figure 1 it seems that all analysis were 1 way ANOVA, comparing the effects of AMPH and MPD in a same hormone treated-group of animals.
It is not understood why in this case they do not compare the effect of the same drug in the CTRL-EV-TP groups, as they have done in the other figures. Comparing CTRL-EV-TP with a 2W ANOVA authors can deduce if the treatment with AMPH/MPD has an impact upon EV-treated rats different from that upon TP-treated-animals. It is very disconcerting and incorrect changing the results expression for no apparent reason.
Deeping in the CPP results, neonatal treatment with hormones (whatever it is) reduces the effect of MPD in males but reduces that of AMPH in females. This result is certainly disturbing and points to a gender difference, dependent on treatment but not on hormone, which should be further evaluated. To go deeper I think it would be very interesting and clarifying to study the CPP at different doses of AMPH or MPD, in neonatal animals treated with EV or TP; and compare the results between males and females with a 3-way NAOVA (drug treatment, neonatal treatment and sex).
In locomotor activity results, authors wrote: “Basal locomotor activity (0-30 min) did not differ between male and female rats following early exposure to sex hormones relative to control rats (Fig. 2A-E) ..... On the other hand, we observed a sex-difference in AMPH-induced locomotor activity between female and male control rats (Fig. 2E)(P=0,0094)”. However, they did not include all the statistical results (F and P values for treatment, sex and interaction. A significant effect of the interaction treatment X sex is required.
Additionally, if there is a gender difference in the in AMPH-induced locomotor activity, it can be deduced that the AMPH dose used is ineffective in males. Therefore, Why has this dose been chosen? What is the scientific reason they chose an ineffective dose of AMPH?
For the results of fig 2B and 2D the authors provide us with a single P. A 2-way ANOVA is required for the variables: treatment (saline, AMPH) and hormones (EV, TP). Only if the interaction and post-hoc test are significant you can say:
“AMPH-induced locomotor activity did not differ between control, EV and TP male rats (Fig. 2B)(F(2, 30) = 0,5718; P=0,5705). However, in EV female rats we observed a significant reduction in AMPH- induced locomotor activity compared to control female rats (Fig. 2D)(F(2, 34) = 4,425; P=0,0196), but not in TP female rat”
Results section 2.3: In vivo brain microdialysis
Line 118: Probably authors have made a mistake mentioning ”amphetamine-induced” instead of methylphenidate-induced.
Results about MPD in males: “W-ANOVA stysted a on significant effect of the interaction, therefore authors cannot deduce that “DA levels induced by systemic MPD administration were lower in EV male rats than control rats”
Section 2.4: In vivo fast scan cyclic voltammetry
Authors must explain the calculation and the significance of TAU.
Data analysis of the results (fig A,B,E,F) using 2W-ANOVA obtained a non significant effect of the interaction. Thus authors cannot perform post-hoc comparisons.
Results section 2.5: DAT mRNA expression in VTA and SN
The statistical analysis performed were 2W-ANOVA. But authors talked about variables: time and neonatal treatment. It’s probably a mistake. Variables must be: sex and neonatal treatment.
For easier reading and understanding, the figures and their corresponding text should be next to each other, not on the next page.
Section 4.7: Statistical analysis
“Two-way ANOVAs followed by Newman–Keuls post hoc tests were performed for all experiments to determine a significant interaction of treatment by sex.”
2W_ANOVA: Post-hoc test only can be performed when the interation between both variables is significant. Additionally, Newman-Keuls test not only requires an interaction significant effect, but also it is not a recommended procedure since and is more likely to reveal significant differences between group means and to commit type I errors by incorrectly rejecting a null hypothesis when it is true. In other words, the Neuman-Keuls is less conservative than Tukey's range test. In ultimate cases you can use Tukey’s honest significant difference (HSD), a post hoc test that does not require a significant interaction between factors and is highly conservative against type I error.
Section 4.4.1: CPP procedure
In the conditioning phases, drug is paired with one compartment and drug vehicle with the other compartment; these compartments and drug/vehicle must be typically counterbalanced across subjects.
How do authors treat animals that during preconditioning express a high bias for one of the compartments? What is the minimum value of preference for one of the compartments during preconditioning to consider that the animal has a high bias?
Round 2
Reviewer 1 Report
The author has answered all my concerns and revised the manuscript accordingly.